# CSDMS Data Components: data-model integration tools for Earth surface processes modeling

Tian Gan[1], Gregory E. Tucker[2,3], Eric W. H. Hutton[1], Mark D. Piper[1], Irina Overeem[1,2], Albert J. Kettner[1], Benjamin Campforts[1], Julia M. Moriarty[1,4], Brianna Undzis[1,4], Ethan Pierce[2], and Lynn McCready[1]

[1] Institute for Arctic and Alpine Research (INSTAAR), University of Colorado Boulder, Boulder, 80309, USA

[2] Department of Geological Sciences, University of Colorado Boulder, Boulder, 80309, USA

[3] Cooperative Institute for Research in Environmental Sciences (CIRES), University of Colorado Boulder, Boulder, 80309, USA

[4] Department of Atmospheric and Oceanic Sciences, University of Colorado Boulder, Boulder, 80309, USA

*Correspondence to*: Tian Gan (gantian127@gmail.com)

**Abstract.** Progress in better understanding and modeling Earth surface systems requires an ongoing integration of data and numerical models. Advances are currently hampered by technical barriers that inhibit finding, accessing, and executing modeling software with related datasets. We propose a design framework for 'Data Components': software packages that provide access to particular research datasets or types of data. Because they use a standard interface based on the Basic Model Interface (BMI), Data Components can function as plug-and-play components within modeling frameworks to facilitate seamless data-model integration. To illustrate the design and potential applications of Data Components and their advantages, we present several case studies in Earth surface processes analysis and modeling. The results demonstrate that the Data Component design provides a consistent and efficient way to access heterogeneous datasets from multiple sources, and to seamlessly integrate them with various models. This design supports the creation of open data-model integration workflows that can be discovered, accessed, and reproduced through online data sharing platforms, which promotes data reuse and improves research transparency and reproducibility.

## 1 Introduction

As the global population increases and infrastructure expands, the need to understand and predict processes at and near the Earth's surface, such as water cycling, landsliding, flooding, permafrost thaw, and coastal change becomes increasingly acute. Progress in understanding and predicting these systems requires an ongoing integration of data and numerical models. Also, given the growing importance of open computational science (Barton et al., 2022; Hall et al., 2022; Lamprecht et al., 2019; Wilkinson et al., 2016), there is a need to overcome technical barriers that inhibit finding, accessing, and operating modeling software tools and related datasets.

To address these challenges, one research focus is the development of modeling frameworks and standards to support model coupling (Hoch et al., 2019; Hutton et al., 2020; Kralisch et al., 2005; Moore & Tindall, 2005; Peckham et al., 2013). These modeling technologies make it easier to integrate diverse models that represent interrelated physical processes to simulate the complex Earth system that drives the movement of water and shapes the planet's surface.

For instance, the Earth System Modeling Framework (ESMF) is a flexible open-source software infrastructure for building and coupling Earth science applications (Hill et al., 2004). The ESMF defines an architecture for composing coupled modeling systems and includes data structures and utilities for developing individual models. Another example is the open World–Earth modeling framework copan:CORE, which is focused on Earth system models with endogenous human societies (Donges et al., 2020) to support the analysis of Earth system dynamics in the

Anthropocene (Verburg et al., 2016).

In the past decade, efforts were also made to design modeling frameworks and tools that improve the reproducibility of data-model integration workflows (Gan et al., 2020b; Hut et al., 2022). For example, the Community Surface Dynamic Modeling System (CSDMS) is an NSF-supported facility that supports and promotes a community of computational modelers of the Earth's surface – the dynamic interface between lithosphere, hydrosphere, cryosphere,

and atmosphere. The CSDMS Workbench is a suite of free and open-source software tools and standards that provide a nimble, plug-and-play environment for model building, coupling, and exploration for Earth surface processes modeling (Tucker et al., 2022). These modeling technologies enable users to write code to create reproducible workflows for coupled model simulations and improve efficiency by reducing the time researchers spend wrestling with idiosyncratic programs and their interfaces. Another example is CyberWater (Chen et al., 2022), a modeling

framework designed to support open data and open model integration for solving environmental and water problems. CyberWater supports direct access to online datasets without tedious work for data preparation, and it includes a generic model agent toolkit to help easily integrate models. This system enables users to create graphical workflows to support data provenance and reproducible computing. The Community Data Models for Earth Predictive Systems (CDEPS https://github.com/ESCOMP/CDEPS) were developed to perform the basic functions of reading external

data files, modifying the datasets and sending the data for Earth system models that are coupled using ESMF. With the development of web technologies and cloud computing, sharing and integrating models across an open web environment also becomes possible. Chen et al. (2020) proposed a conceptual framework for open web-distributed integrated modeling and simulation, which is intended to enhance the use of existing resources and help people in different locations and from various research fields to perform comprehensive modeling tasks collaboratively.

In addition, there are several organizations that provide the scientific community with online platforms for sharing research datasets, models, and tools to improve the findability, accessibility, interoperability, and reusability (the FAIR principles) of digital research objects (Lamprecht et al., 2019; Wilkinson et al., 2016; Chue Hong et al., 2021). For instance, CSDMS maintains an online Model Repository (Tucker et al., 2022) that catalogs over 400 open-source models and tools, ranging from individual subroutines to large and sophisticated integrated models. The Model

Repository now includes about 20,000 references to literature describing these models and their applications, giving prospective model users efficient access to information about how various codes have evolved and are being used. Similarly, the Network for Computational Modeling in Social and Ecological Sciences (CoMSES Net) provides an

extensive Model Library of codes used in social and ecological sciences, together with a curated database of over 7,500 publications (Janssen et al., 2008). For water-related sciences, HydroShare (Gan et al., 2020a; Horsburgh et al., 2015) provides a web-based hydrologic information system to share and publish data and models in various formats that are created by individual researchers and research groups. This platform enables researchers to collaborate and work in an online environment to enhance research and education and improve the reproducibility of the research results. Geoscience Cyberinfrastructure for Open Discovery in the Earth Sciences (GeoCODES https://www.earthcube.org/geocodes) is another effort aiming to improve the discovery and access of research datasets and tools. GeoCODES provides a data standard and a set of tools to expose, index, and query datasets across repositories.

Although many modeling technologies and cyberinfrastructures are available to support open data and model integration, challenges still exist. For example, rapid advances in observational data using remote sensing and other technologies have brought about a data revolution, and with it the potential for substantial improvement in our ability to understand and predict a diverse array of Earth systems. However, the majority of model frameworks and systems lack an effective mechanism to easily access datasets from a variety of sources and couple them with the models. Although some model frameworks and systems can use web services to access various datasets and provide them as model inputs, the problem remains that the data access and preparation methods tend to be developed around specific models or model frameworks, and the corresponding details are either hidden behind a graphical user interface (GUI) or provided with scripts that offer only limited options for the users. It is challenging for researchers to understand or modify the data access or preparation methods for their research needs, which inhibits the research transparency and impedes flexibility. Moreover, it is often difficult to reuse data access methods for different modeling frameworks, which leads to redundant programming efforts.

To address these challenges, we present the design and development of the CSDMS Data Components. This design is built on the model coupling technologies from the CSDMS Workbench to enable data access through plug-and-play components, and thereby integrate datasets with models. This design aims to provide a consistent way of using datasets across multiple sources to better facilitate the integration of heterogeneous datasets with models for Earth surface processes. This design also supports creating data-model integration workflows that can include detailed data access and preparation steps, and can be shared and executed on cloud platforms to enable the geoscience community to discover, access, and reproduce computational modeling research. In addition, the proposed design provides the flexibility to couple Data Components under different modeling frameworks with minimal coding effort.

In this paper, Section 2 presents the background for the CSDMS model coupling technologies and the Data Component design. Section 3 presents case studies for Data Component implementation and their use cases for Earth surface processes modeling. Section 4 provides the summary and conclusions.

## 2 Methods

### 2.1 CSDMS Workbench

Since the Data Component design is based on the CSDMS Workbench, we will first introduce its underlying modeling technologies, including the Basic Model Interface (BMI), Babelizer, Python Modeling Toolkit (pymt), and Landlab.

**BMI** is an interface specification that identifies a minimal set of functions necessary for dynamic coupling of data to models or models to other models. The BMI concept was first introduced as a foundational technology for the CSDMS model coupling framework (Peckham et al., 2013). The current version of BMI updated the original design with new functions for describing variables and for working with structured and unstructured grids (Hutton et al., 2020; Tucker et al., 2022). BMI is a language-neutral standard that is defined using the Scientific Interface Definition Language

(SIDL) (Epperly et al., 2011). CSDMS has defined language-specific BMI specifications for Python, C, C++, Java, and Fortran, which are the most commonly used languages for Earth system models; other groups have created specifications for additional languages such as Julia and JavaScript. BMI is designed to be framework agnostic, and to be as easy as possible for a developer to implement. This means that a component that exposes a BMI can be incorporated into any framework and does not need to be modified to add any BMI-specific dependencies into the

component. Several modeling frameworks that support model coupling (Hoch et al., 2019; Hut et al., 2022) have been built upon the BMI. Two such BMI-capable frameworks, pymt and Landlab, are described below.

**Babelizer** is a command-line utility that creates a Python-importable package to present a BMI component as a Python class (Hutton et al., 2022). Language interoperability is critical to a model coupling framework that brings together models written in a range of programming languages. One of the approaches to tackle this challenge is to use a hub

language, through which other languages will communicate, and to build bridges from each supported language to the hub language. CSMDS adopted this approach for the Babelizer and chose Python as the hub language. The Babelizer helps streamline the process of bringing a BMI component written in C, C++, or Fortran into Python and it is easily extensible to support other languages.

**pymt** is a Python-based model coupling framework that provides a set of utilities for running and coupling BMI

components (for both models and data) (Tucker et al., 2022). This model coupling framework consists of three major pieces. The first is a collection of legacy models that represent a diverse set of environmental systems. Models in the pymt collection are written in a variety of languages (e.g., C, C++, and Fortran), but are wrapped with a BMI as a common interface. The second piece is a wrapper for BMI components that augments them with additional capabilities, such as memory management, unit conversion utilities, and grid mappers. The third piece is a set of

utilities for performing common model-coupling tasks, which includes the grid interpolation via the ESMF grid mapping engine (used when models or data operate on different grids) (ESMF Reference Manual for Fortran, 2023), time interpolation (used when models or data operate on different intervals), unit conversion through the UDUNITS package (https://www.unidata.ucar.edu/software/udunits/), and a coupling orchestrator that organizes the time stepping of a set of components.

**Landlab** is a toolbox for building new components within a Python-based (BMI compatible) modeling framework (Hobley et al., 2017; Barnhart et al., 2020). Landlab includes three major elements that speed up model development

and analysis. The first is a gridding engine that allows model developers to create a grid in as little as a single line of code, and that provides users a choice of grid type (e.g., a structured rectilinear grid versus an unstructured mesh). The second piece is a growing collection of modularized components that model single physical processes (e.g., overland flow or hillslope process) or perform an analysis operation (e.g., calculate the down-slope direction at each grid cell in a digital elevation model). The third element is a library of utilities for common operations such as file input and output that includes standard formats such as NetCDF, ESRI ASCII, and Legacy VTK. The Landlab library provides components that can be brought into other frameworks and, additionally, be automatically wrapped with BMI, allowing them to operate within BMI-friendly systems such as pymt.

## 2.2 Data Component Design

A Data Component is a dataset that is wrapped with a BMI. When a model is equipped with a BMI, we refer to it as 'Model Component'. Model Components make models easier to learn and to couple with other models because of the similarity in control and query functions. Similarly, by wrapping datasets with BMI functions, we provide a consistent way to access various types of datasets without considering their specific file formats and making them easier to integrate with Model Components. Thus, the Data Component extends the application of BMI from models to datasets. With BMI, Model and Data Components use the same functions to initialize the component, control its execution (e.g., advance a model or dataset in time), and access variables, grid, and/or time information. Both applications use configuration files to specify the detailed information needed to initialize component instances. Table 1 lists the example BMI functions for each category. (Note that not all BMI functions are necessarily relevant for every Data Component. For example, for a dataset that lacks time-stamped data, the time-related functions would not be needed, and would simply return null values.)

The specifications for the Data Components are designed to meet the following requirements:

- Access datasets from either a remote server or a local file system. Remote servers provide web services and/or a corresponding application programming interface (API) to support programmatic data access.

- Use the same data structure to manage datasets stored in different file formats (e.g., CSV, GeoTIFF, or NetCDF) and grid types (e.g., 1D, 2D or 3D array) for time series, raster, or multidimensional space-time data.

- Use open-source tools and standards for Data Component implementation and avoid dependencies on proprietary software.

- Expose a BMI so that Data Components can be used within different modeling frameworks without the need to modify their implementation.

The Data Component design is based on the CSDMS Workbench and includes two major elements (Fig. 1). The first element is the BMI component which can be implemented as a Python package to download the datasets and wrap them with BMI functions (Table 1). This package includes an API, which can be implemented as a Python class to access and retrieve the datasets from a remote server. The corresponding command line interface (CLI) can also be included, which allows users to download datasets through shell commands. The datasets can be cached locally and

loaded as an xarray object (Hoyer and Hamman, 2017) to satisfy the need for using the same data structure to manage datasets in various formats and grid types. The second element is a Babelized component, which is a Python package created by the Babelizer. This Babelized component converts the BMI component into a plug-and-play component for the modeling frameworks (e.g., pymt). It can also help import the BMI components that are implemented in other languages as a Python class, so that they can communicate with each other using the hub language (Python). For the second element, the developer only needs to provide metadata describing the BMI component through a toml-format file. The Babelizer will then use the metadata to construct a Python package, which is almost completely autogenerated (Hutton et al., 2022). This design minimizes the effort of using the Data Component within different modeling frameworks, because there is no need to change the BMI implementation and one only needs the Babelizer and the required metadata to create a component for any relevant modeling framework. Generally, the BMI component is the fundamental essence of the Data Component, while the Babelized component represents the Data Component for a specific modeling framework.

**Table 1 List of BMI functions shared by Model and Data Components.**

| Function Category | Function Name | Description |
|---|---|---|
| component control | initialize | Perform startup tasks for the component. |
| | update | Advance component state by one time step. |
| | finalize | Perform post execution tasks for the component. |
| component information | get_component_name | Name of the component. |
| | get_output_names | List of a component's output variables. |
| | get_output_item_count | Number of a component's output variables. |
| variable information | get_var_grid | Get the grid identifier for a variable. |
| | get_var_units | Get the units of a variable. |
| | get_var_type | Get the data type of a variable. |
| | get_var_location | Get the grid element type of a variable. |
| time information | get_current_time | Current time of the component. |
| | get_time_units | Time units used in the component. |
| | get_time_step | Time step used in the component. |
| grid information | get_grid_type | Get the grid type as a string. |
| | get_grid_shape | Get the dimensions of a computational grid |
| | get_grid_spacing | Get the spacing between grid nodes. |
| variable getter and setter | get_value | Get current values for a variable. |
| | set_value | Set current values for a variable. |

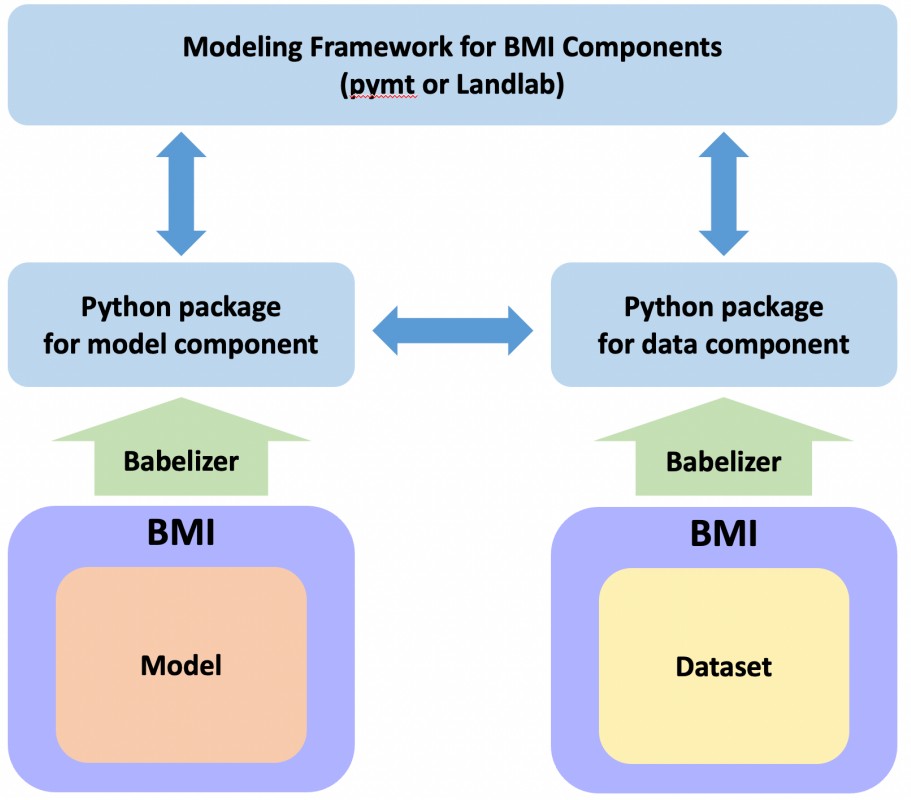

**Figure 1: Relationship between datasets, models, and the CSDMS Workbench tools**.

To test the Data Component design, we conducted case studies by implementing several Data Components and creating use cases for Earth surface processes modeling and analysis. These datasets are from multiple data providers and in various file formats and grid types. The use cases are data-model integration workflows created as Jupyter Notebooks and shared in HydroShare. We also installed the CSDMS Workbench tools on the CUAHSI JupyterHub (https://help.hydroshare.org/apps/CUAHSI-JupyterHub/) and the CSDMS JupyterHub (https://csdms.colorado.edu/wiki/JupyterHub). This enables users to discover and access these use cases from HydroShare and use the CUAHSI or CSDMS JupyterHub to reproduce the modeling workflows without the need of software installation and data download on the local computers. Moreover, users can also use the environment files which are prepared for these use case Jupyter Notebooks to build local virtual environments and run them. Detailed results and discussion are presented in the next section.

## 3 Case Studies

### 3.1 Data Components

We implemented multiple Data Components to demonstrate the access to widely used datasets for Earth surface processes modeling. To illustrate the broad applicability of Data Components, these examples cover several domains: hydrology, topography, soil, meteorology, and oceanography. The data types span the categories of time series, geographic raster, and multidimensional space-time data. Here we provide an overview of each Data Component.

**The NWIS Data Component** (Gan, 2023c) is implemented to access time series of hydrological data from the US
Geological Survey's National Water Information System (NWIS https://waterdata.usgs.gov/nwis). NWIS provides a RESTful (Representational State Transfer) web service to access current and historical water-resources datasets across the US, such as discharge, gage height, and water temperature. REST web services allow users to access data using a Uniform Resource Identifier (URI), which distinguishes one resource from another (e.g., links on the web). Our NWIS Data Component can download the time series for instantaneous and daily values from NWIS using the 'dataretrieval'
Python package (Hodson et al., 2023), which is a Python client for the REST web services of NWIS. This Data Component needs a configuration file that specifies USGS site number, start and end time, USGS variable code, and output file name. Each Data Component supports storage of the dataset in a NetCDF file which can include time series for multiple variables at multiple USGS sites. The time values are stored in a format by following the Climate and Forecast (CF) metadata conventions (http://cfconventions.org/).

**The Topography Data Component** (Piper, 2023) fetches global terrain elevation raster data from OpenTopography (https://opentopography.org/), an NSF-supported facility that provides access to many different types of topography data, alongside related tools and resources. OpenTopography provides REST web services to retrieve raster datasets such as NASA Shuttle Radar Topography Mission (SRTM) and JAXA Advanced Land Observing Satellite (ALOS) global data (Tadono et al., 2014; Farr et al., 2007). These REST web services were used to implement an API and a
CLI in the Topography Data Component for downloading these datasets. Dataset type, latitude-longitude bounding box, and the desired output file format can be specified with arguments to this Data Component or through a configuration file. As of this writing, users are required to apply for an API key from OpenTopography to be authorized for data access, which helps OpenTopography monitor and understand the usage of the REST web services and to provide a more stable and secure user experience. For this data component, we implemented a utility function to help
access the API key on local computers to simplify the process for data access authorization.

**The SoilGrids Data Component** (Gan, 2023d) provides access to global gridded soil data from SoilGrids (https://www.isric.org/explore/soilgrids), a system for global digital soil mapping that uses machine learning methods to map the spatial distribution of soil properties (Poggio et al., 2021; Hengl et al., 2017). The SoilGrids system provides web coverage services (WCS) to help users obtain a subset of the soil maps as raster datasets for soil properties such
as bulk density, clay content, and soil organic carbon content. The WCS were used to implement the API and CLI in the SoilGrids Data Component to download the desired soil datasets and store them in a local GeoTIFF file. This Data Component requires a configuration file that includes the information for the map service name, bounding box, coordinate system, grid resolution, and other parameters. Fig. 2 shows the example scripts that use the API and the

Babelized component (e.g., pymt component) to access and visualize the same soil property dataset from SoilGrids

system.

**The ERA5 Data Component** (Gan, 2023b) accesses the ERA5 climate dataset, which is available in the Copernicus Climate Data Store (CDS https://cds.climate.copernicus.eu/). ERA5 refers to European Centre for Medium-Range Weather Forecasts (ECMWF) reanalysis 5, which includes multidimensional space-time datasets produced using data assimilation and model forecasts for the global climate and weather for the period from the 1950s to near real time.

The ERA5 Data Component downloads data using the 'cdsapi' Python package, which is the API for retrieving datasets from the CDS platform. This Data Component requires a configuration file that includes information for data variables, time period, latitude-longitude bounding box, grid resolution, and other parameters. Each ERA5 Data Component supports storing the datasets in a NetCDF file, which can contain multiple variables for a given bounding box area. Similar to the Topography Data Component, users are required to apply for API keys from the CDS platform

to be authorized for data access. We implemented a utility function to help generate the API key files on the local computers for data access authorization.

**The WAVEWATCH III Data Component** (Hutton, 2023) retrieves data from the global wave datasets (https://polar.ncep.noaa.gov/waves/product_table.shtml) that are generated with the WAVEWATCH III model (Booij et al., 1999). These model outputs are multidimensional space-time datasets for wave height, period, direction, and

other attributes. The WAVEWATCH III Data Component includes an API and a CLI, which use web services to download the 30-year wave hindcast (Phase 1 and 2) and the production hindcast (single grid and multigrid) datasets and store them as GRIB-formatted files. This Data Component requires a configuration file that includes the information for time, grid type, and data source.

The Data Components for the time-varying datasets such as NWIS and ERA5 retrieve the datasets once and save them

in a file when the 'initialize' method is used. If a user runs the same Data Component with an identical configuration file multiple times on the same machine, the data will be downloaded only during the first instance to prevent redundant download processes. Aside from the Data Components presented here, we also implemented other Data Components. A full list of them can be found at https://csdms.colorado.edu/wiki/DataComponents.

```python
import matplotlib.pyplot as plt
from soilgrids import SoilGrids

# get data from SoilGrids
soil_grids = SoilGrids()
data = soil_grids.get_coverage_data(service_id='clay',
                                    coverage_id='clay_0-5cm_mean',
                                    west=-1784000, south=1356000,
                                    east=-1140000, north=1863000,
                                    crs='urn:ogc:def:crs:EPSG::152160',
                                    output='demo.tif')
# plot data
data.plot(figsize=(10,6))
plt.ylabel('Y (m)', fontsize=12)
plt.xlabel('X (m)', fontsize=12)
plt.title('Mean clay content (g/kg) at 0-5cm soil depth in Senegal')
```

**(a)**


```python
import matplotlib.pyplot as plt
from pymt.models import SoilData

# initiate a data component
data_comp = SoilData()
data_comp.initialize('config.yaml')

# get variable and grid metadata
var_name = data_comp.output_var_names[0]
var_grid = data_comp.var_grid(var_name)
grid_shape = data_comp.grid_shape(var_grid)
grid_spacing = data_comp.grid_spacing(var_grid)
grid_origin = data_comp.grid_origin(var_grid)

# get variable data
data = data_comp.get_value(var_name)
data_2D = data.reshape(grid_shape)

# get X, Y extent for plot
min_y, min_x = grid_origin
max_y = min_y + grid_spacing[0]*(grid_shape[0]-1)
max_x = min_x + grid_spacing[1]*(grid_shape[1]-1)
dy = grid_spacing[0]/2
dx = grid_spacing[1]/2
extent = [min_x - dx, max_x + dx, min_y - dy, max_y + dy]

# plot data
fig, ax = plt.subplots(figsize=(10,6))
im = ax.imshow(data_2D, extent=extent)
fig.colorbar(im)
plt.ylabel('Y (m)', fontsize=12)
plt.xlabel('X (m)', fontsize=12)
plt.title('Mean clay content (g/kg) at 0-5cm soil depth in Senegal')
```

**(b)**

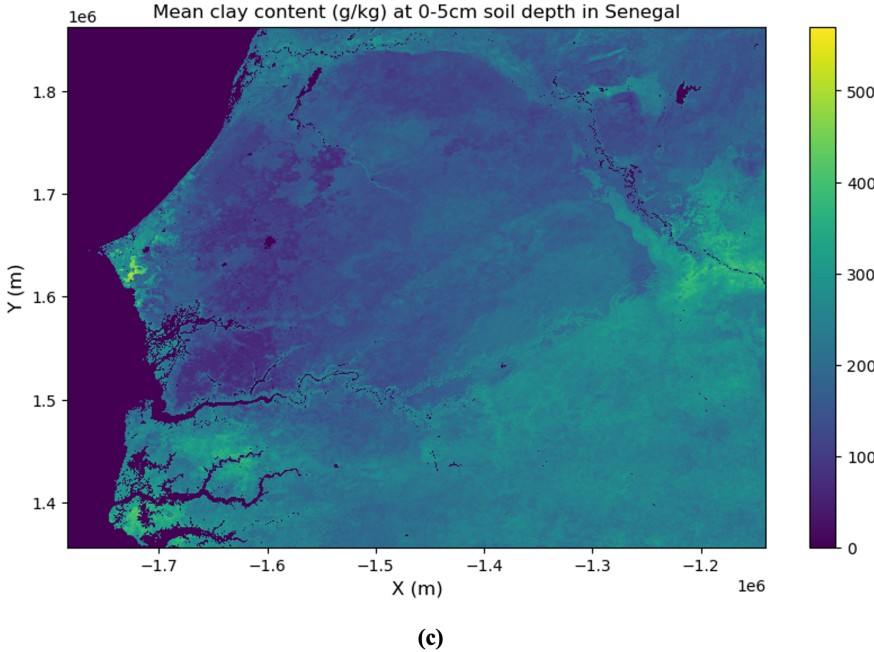

**(c)**

**Figure 2: Example scripts that use the API (a) and the pymt component (b) of the SoilGrids Data Component to access and visualize the soil property dataset (c).**

### 3.2 Use Cases

Here we present use cases that cover a variety of topics, including landslide susceptibility mapping, modeling of overland flow in a wildfire-impacted catchment, permafrost landscape processes, and wave power analysis (Gan, 2023a). These use cases serve as illustrative examples for the application of Data Components across varied domains. We will focus on describing the modeling workflows and discussing multiple ways of using the Data Components presented in these use cases, rather than new research findings and analysis details. The data-model integration workflows for these use cases can be discovered, accessed, and reproduced on the HydroShare platform or the CSDMS web portal.

### 3.2.1 Landsliding

Landslides are a dominant source of sediments in mountain regions (Broeckx et al., 2020). Landslides cause thousands of casualties annually, together with expensive damage to infrastructure (Haque et al., 2016; Petley, 2012). Landslides are also point sources of sediment in riverine systems, altering stream geomorphology (Benda and Dunne, 1997), potentially creating landslide dams and subsequent failures (Costa and Schuster, 1988), altering ecosystem functioning (May et al., 2009), and increasing downstream flood risk (Fan et al., 2019). Our example use case focuses on Puerto Rico, where a combination of steep terrain and heavy rainfall from hurricanes makes landslides a common occurrence. For example, Hurricane Maria made its landfall on September 20th, 2017 and triggered more than 40,000 landslides

(Bessette-Kirton et al., 2019). In this use case, we chose a study area that had a high concentration of landslides during Hurricane Maria. We used several Data Components to generate landslide susceptibility maps in this region.

We adopted the method of Strauch et al. (2018) to calculate landslide susceptibility using a factor-of-safety approach. This method requires data for soil depth, terrain slope, and subsurface flow depth. To prepare those inputs, we used the Topography and ERA5 Data Components to access terrain elevation, soil moisture content, and precipitation datasets. We also retrieved the soil depth-to-bedrock dataset from the SoilGrids system. Terrain slope was derived by combining the Topography Data Component with a Landlab RasterModelGrid object to calculate the slope angle.

Subsurface flow depth was calculated by using the soil depth and soil moisture content datasets. The precipitation data is not used for input preparation but rather for visualization purposes to show the water input conditions in the study area. Since these datasets are in different grid resolutions, we performed data regridding to interpolate the soils and precipitation data to the same resolution as the SRTM terrain elevation data (~90 by 90 m per grid cell). Using these inputs, we looped through 48 one-hour time steps (for Sept 20-21, 2017, the time period over which Hurricane Maria

made landfall) to generate hourly results. The hourly maps were used to create an animation that shows the changes in landslide susceptibility and subsurface flow depth over the two-day period. The time series of mean total precipitation and soil moisture content at four soil layers (layer 1: 0 – 7 cm, layer 2: 7 – 28 cm, layer 3: 28 – 100 cm, layer 4: 100 – 289 cm) for the study area are also shown in the results. Fig. 3 shows the input terrain elevation and slope maps, and Fig. 4 shows an example output. When the precipitation reached its peak, soil layers 1 and 2 responded

quickly and reached high soil moisture content, while layer 3 responded with a time lag and layer 4 kept with a low value. The areas where the landslide susceptibility increased most correspond to the areas that have high slope angle and a greater increase in subsurface flow. Landslide susceptibility mapping is an important approach for evaluating the likelihood of a landslide occurring in an area, which provides critical support to reduce disaster loss. This use case highlights the value of Data Components for recreating near-real time landslide susceptibility maps in regions prone

to the landslide hazards, or to do first-order exploratory simulations in response to a large landsliding event anywhere in the world.

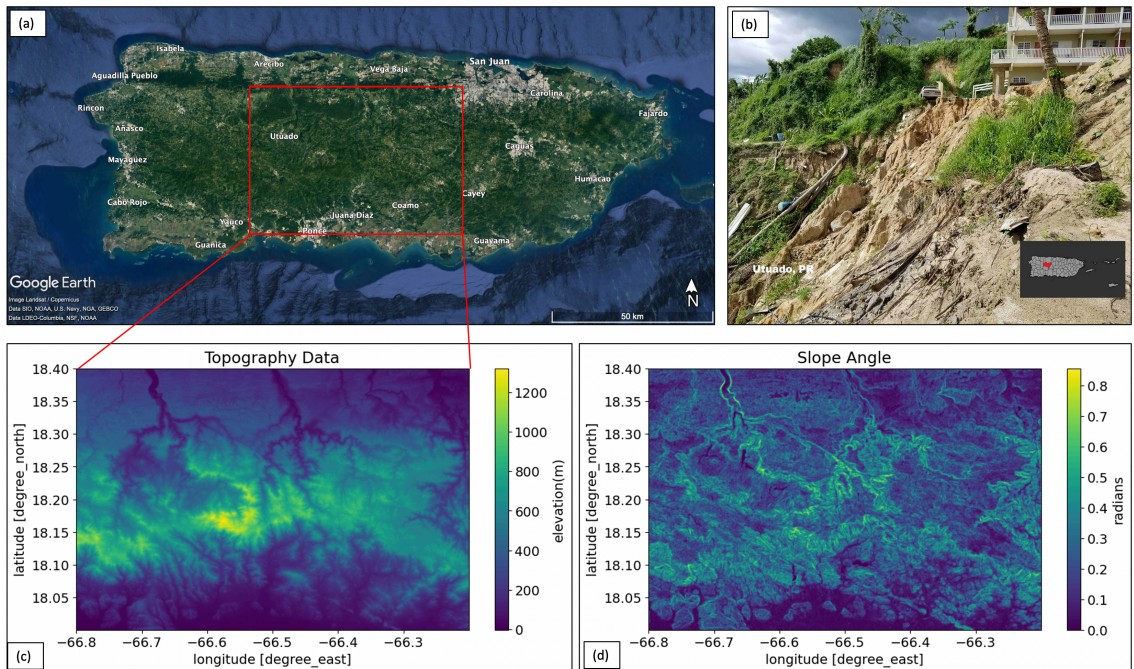

**Figure 3: The study area in Puerto Rico. Panel (a) shows the bounding box of the study area; (b) shows a field photo of a landslide in the study area after Hurricane Maria (source from NOAA weather service https://www.weather.gov/sju/maria2017); (c) shows the terrain elevation data; (d) shows the calculated slope angle using the Landlab RasterModelGrid component.**

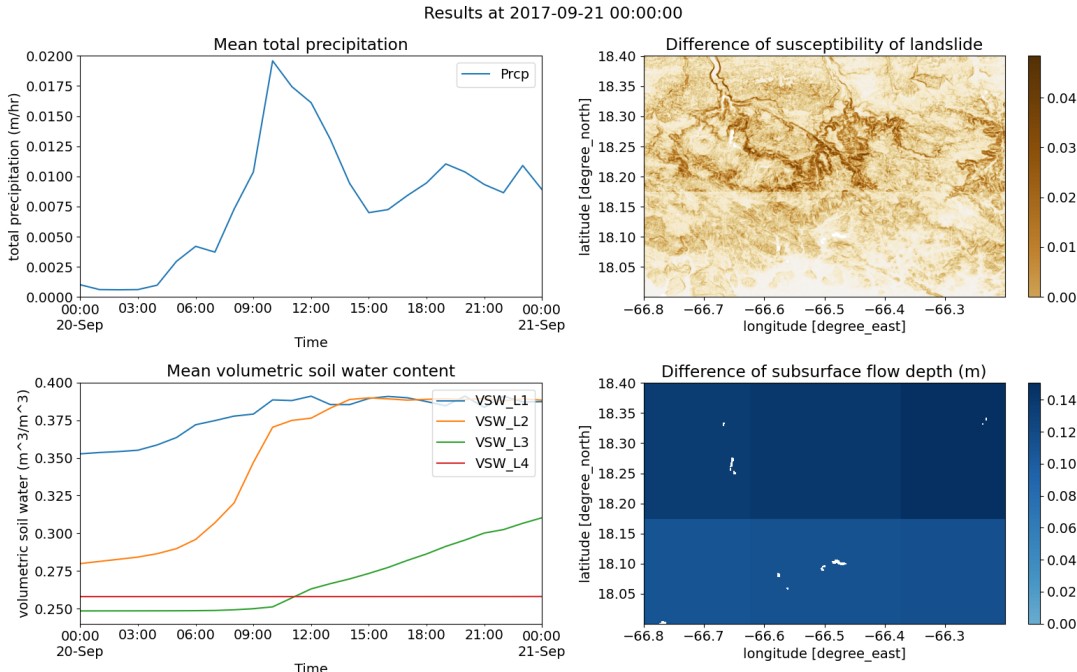

**Figure 4: Example result for the study area in Puerto Rico. The left panel shows the mean total precipitation and the mean volumetric soil water content at four soil layers; the right panel shows the difference of landslide**

**susceptibility and the subsurface flow depth between the first (2017-09-20 00:00) and the current (2017-09-21 00:00) time step.**

**3.2.2 Rainfall-runoff modeling in wildfire-affected watersheds**

Storm runoff occurs when saturated soil cannot absorb additional water (saturation-excess mechanism) or when the rate of water input on the land surface is higher than the infiltration rate (infiltration-excess mechanism). The generation of runoff is mainly impacted by the intensity of rainfall and the landscape surface characteristics such as vegetation density (surface roughness), antecedent moisture condition, and slope. In particular, after a destructive wildfire burns away plants and trees and affects the soil to alter the site characteristics (Shakesby and Doerr, 2006),

heavy rain can cause substantial overland flow and potentially trigger debris flows (Malvar et al., 2011; Cannon et al., 1998). In the western US, wildfires are already increasing in size and frequency, and the frequency and intensity of post-fire overland flow are likely to increase even further in the future (Beeson et al., 2001; Halofsky et al., 2020; Abatzoglou et al., 2021). Thus, it is important to simulate overland flow processes to study the hydrologic responses of burned watersheds. In this use case, we performed a rainfall-runoff simulation for the watershed of Geer Canyon

in the Colorado Front Range (USA), northwest of the city of Boulder (Fig. 5a, 5b). This watershed was impacted by the CalWood Fire, which occurred in 2020 and burned more than 4,000 hectares.

In this use case, we used the Topography Data Component to retrieve terrain elevation data for the study area (Fig. 5c). We performed a watershed delineation (Fig. 5d) by coupling this Data Component with Landlab components, specifically FlowAccumulator and ChannelProfiler (Barnhart et al., 2020). Then we used the watershed terrain

elevation as input for a model of rainfall and runoff using Landlab's OverlandFlow component (Adams et al., 2017). The model run time is set as 200 minutes with the first 10 minutes assigned a constant rainfall intensity (59.2 mm/hr) based on the meteorological observations on June 25, 2021, the summer after the CalWood fire occurred. This simulation created a discharge time series plot at the watershed outlet and a map of the surface water depth over the watershed at each 30-second time step (Fig. 6). Finally, an animation was made to show the overland flow process

during the simulation time. This use case demonstrates the ability to couple a Data Component with Landlab components for post-fire overland flow simulation and for exploring a watershed storm response after fire events. This modeling workflow can be applied to perform experiments by adjusting the model parameters and inputs (e.g., surface roughness, infiltration rate, rain intensity) to evaluate the impact of wildfire on hydrologic responses for watersheds more generally.


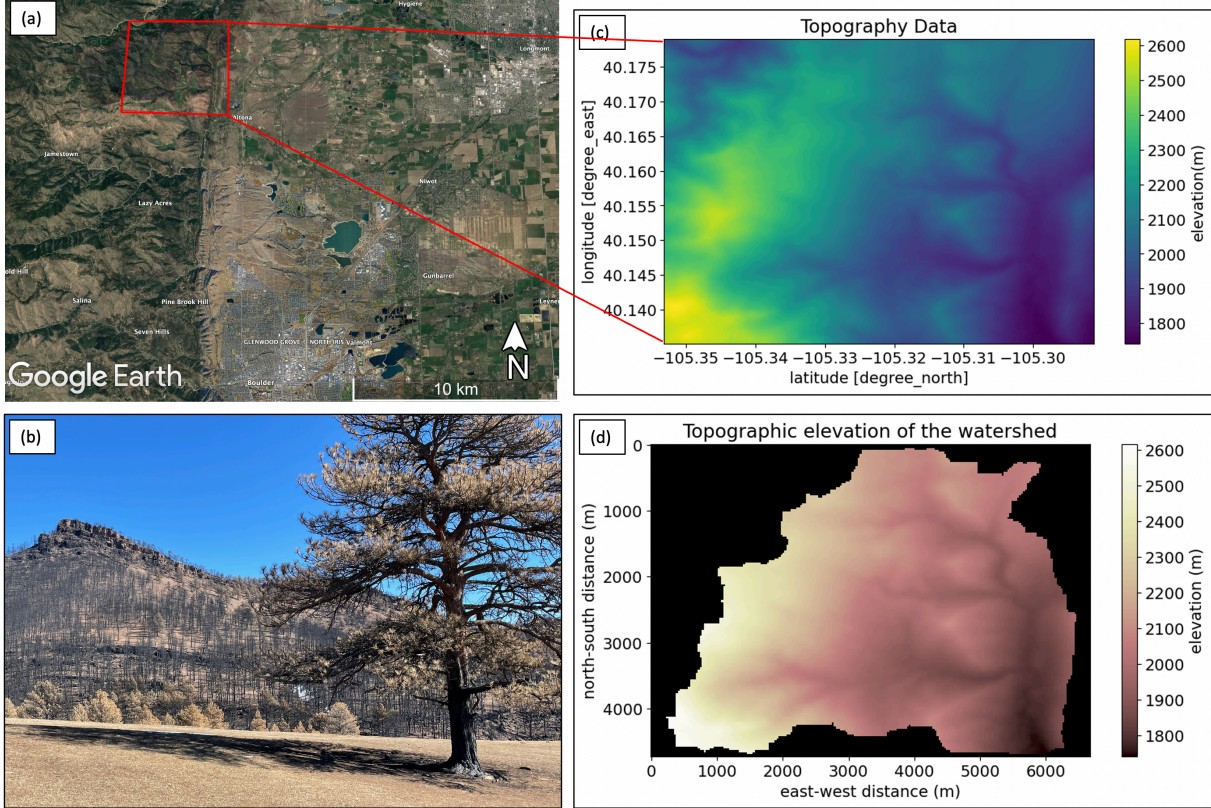

**Figure 5: The watershed of Geer Canyon. Panel (a) shows the bounding box of the study area; (b) shows field photo of the burned study area in March 2021; (c) shows the terrain elevation data; (d) shows the watershed delineation result using the Landlab FlowAccumulator and ChannelProfiler components.**

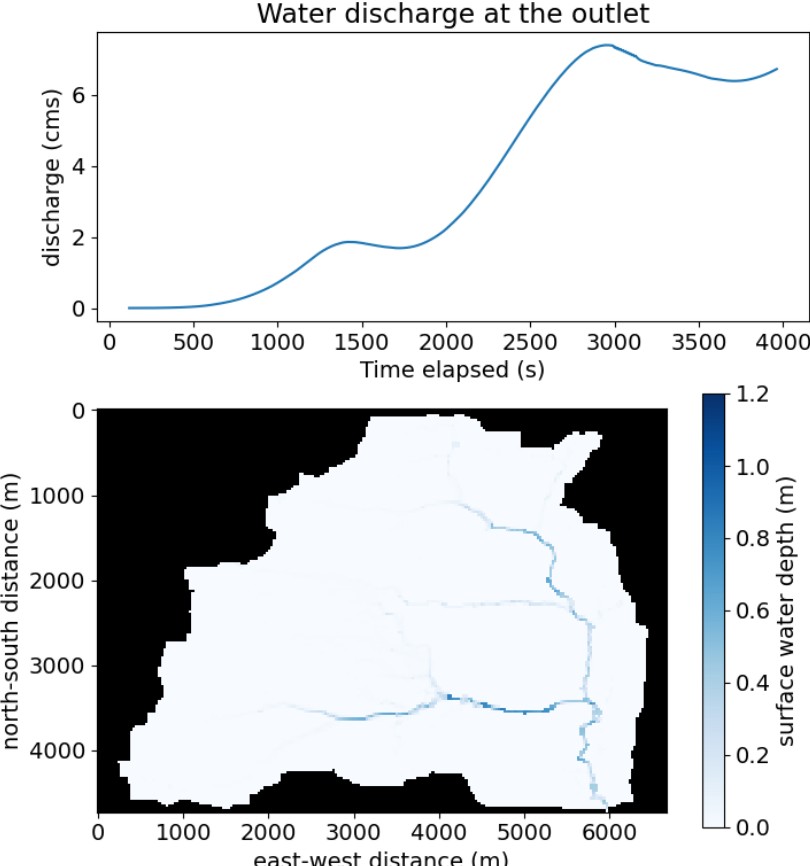

**Figure 6: Example result of discharge and surface water depth from the Landlab OverlandFlow component for the watershed of Geer Canyon. This watershed is not gauged at its outlet but flows overbanked Geer Creek and spilled over the adjacent road in the June 25th rain event (personal comment Boulder Open Space).**

### 3.2.3 Permafrost thaw and hillslope diffusion

Permafrost is defined as rock or soil that remains below 0°C for two or more consecutive years. Nearly a quarter of soils in the Northern Hemisphere are permafrost-affected (Zhang et al., 2008). Due to the ongoing impact of global warming, more permafrost is thawing as temperatures rise above freezing. This results in geologic hazards such as landslides, ground subsidence, erosion, and other severe surface distortions  (Lawrence and Slater, 2005; Nelson et al., 2001; Patton et al., 2019). Research for the future transformation of the permafrost in a changing climate becomes vital to reduce the negative impact of thawing permafrost on, for example, coastal erosion and infrastructure (e.g., roads and houses), and to assess the potential for the release of soil carbon to the atmosphere. In this use case, we applied the Kudryavtsev model (Anisimov et al., 1997; Kudryavtsev et al., 1977) for a study area in Alaska to evaluate

the impact of the warming climate on the thickness of the active layer of permafrost. Additionally, we applied the Kudryavtsev model output, the active layer thickness, as the input for a hillslope soil transport model to predict

hillslope evolution in the Eight Mile Lake area, just south of Denali National Park.

The Kudryavtsev model includes thermodynamic processes that provide a steady-state solution under the assumption of a sinusoidal air temperature forcing to predict the annual active layer thickness and snow surface temperature. This model has been implemented as a pymt Model Component, for which the major inputs include annual mean temperature, amplitude of annual temperature variation, and snow cover depth. We obtained monthly mean air

temperature, snow density, and snow water equivalent data using the ERA5 Data Component, and further processed these quantities to provide model inputs. To evaluate the impact of a warming climate, we prepared two sets of inputs—for 1980-1989 and 2010-2019, respectively—to compare their corresponding model outputs. Fig. 7 shows the model input time series and Fig. 8 shows the model output of the annual active layer thickness. These plots show that the annual mean temperature tends to increase while the temperature amplitude and snow cover depth became

lower in 2010-2019 than in 1980-1989. However, the warming and drying climate didn't lead to a significant change in the active layer thickness. We conducted model experiments to find out the reason. We first calculated the 10-year average of annual mean temperature, amplitude of annual temperature variation, and snow cover depth for 1980-1989 and 2010-2019. Then we used these inputs to conduct two model runs for those periods. The model result for 1980-1989 will be taken as the 'base' experiment for comparison. We then conducted 3 model runs of which each

experiment used two inputs from the 10-year average for 1980-1989 and one for 2010-2019. The results showed that it can lead to an increase of the active layer thickness by only increasing the annual temperature. But if the snow thickness decreases, its insulating capacity in mid and late winter will also decrease, and as a result the active layer will actually become thinner. Therefore, warming temperature and decreasing snow thickness can act in opposing directions and thereby minimize changes in active layer thickness. This phenomenon was also observed with field

datasets and studied by several researchers at other study sites (Garnello et al., 2021; Zhang, 2005).

To examine the potential impact of active layer thickening on soil transport, we implemented a simple model of hillslope evolution using the Landlab DepthDependentDiffuser component to simulate the modification of topography by thaw-enhanced soil creep. The Topography Data Component was used to prepare the terrain elevation input (Fig. 9), and the active layer thickness for 2010-2019 was used as the soil depth input to the hillslope evolution model. We

performed a model simulation representing 1,000 years of geomorphic evolution and made an animation to show the changes in terrain elevation. This use case provides an example of coupling Data Components with both pymt and Landlab Model Components, which shows the flexibility of integrating Data Components with multiple modeling frameworks to simulate interrelated landscape surface processes.

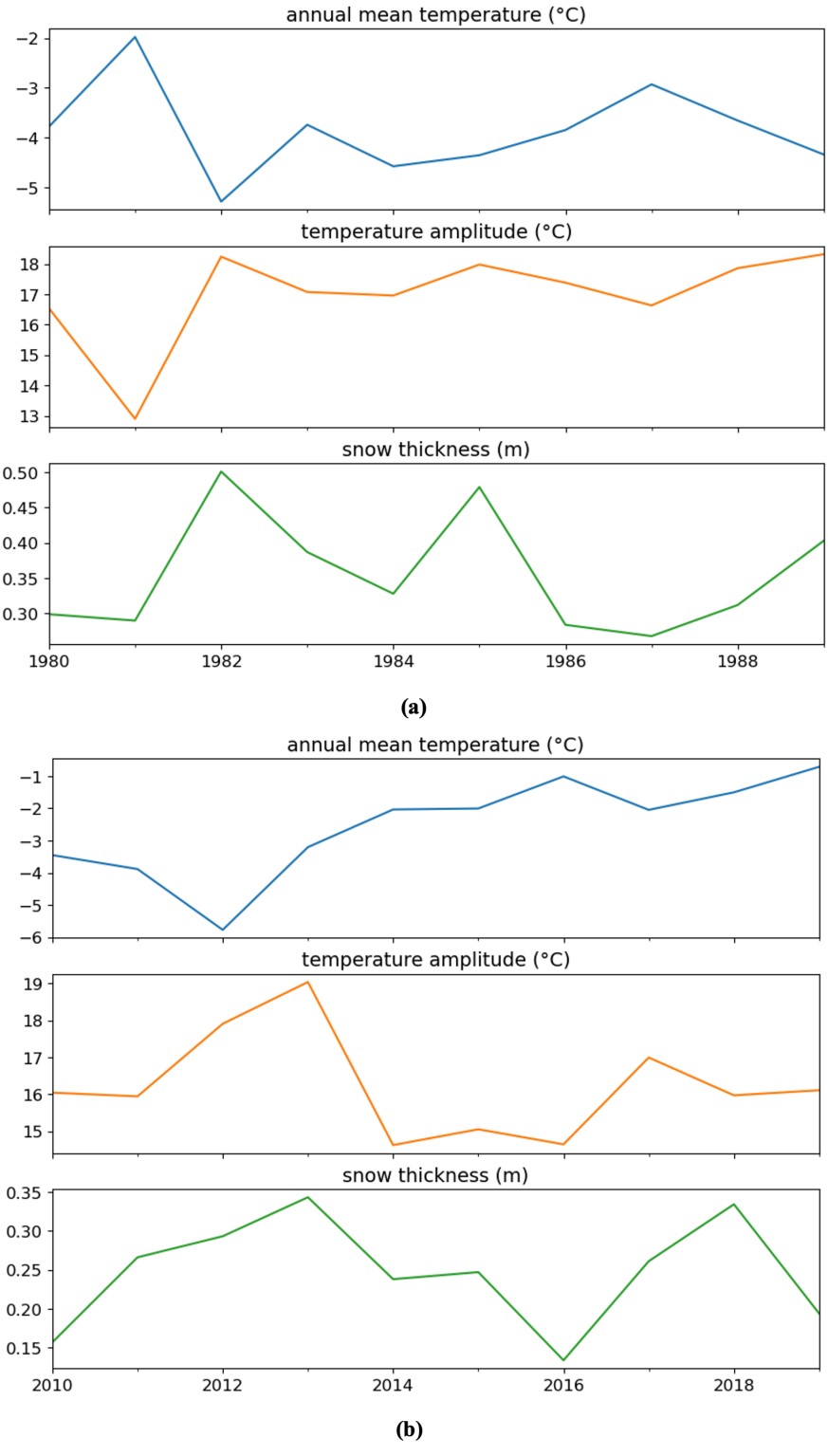

**Figure 7: Temperature and snow inputs of the Kudryavtsev model for the Eight Mile Lake area. Panel (a) for 1980-1989 and (b) for 2010-2019.**


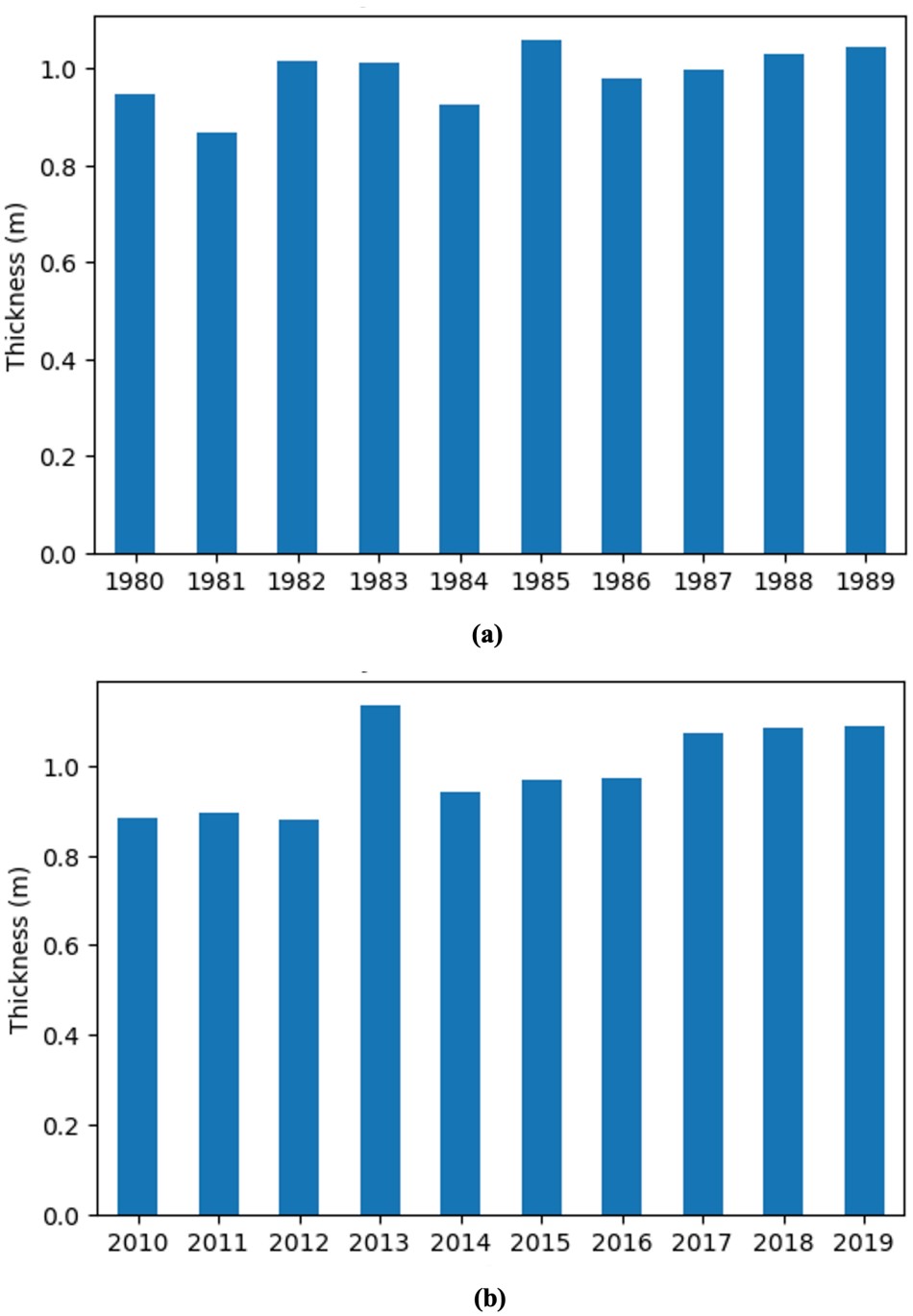

**(a)**

**(b)**


**Figure 8: Active layer thickness results of the Kudryavtsev model for the Eight Mile Lake area. Panel (a) for 1980-1989 and (b) for 2010-2019.**

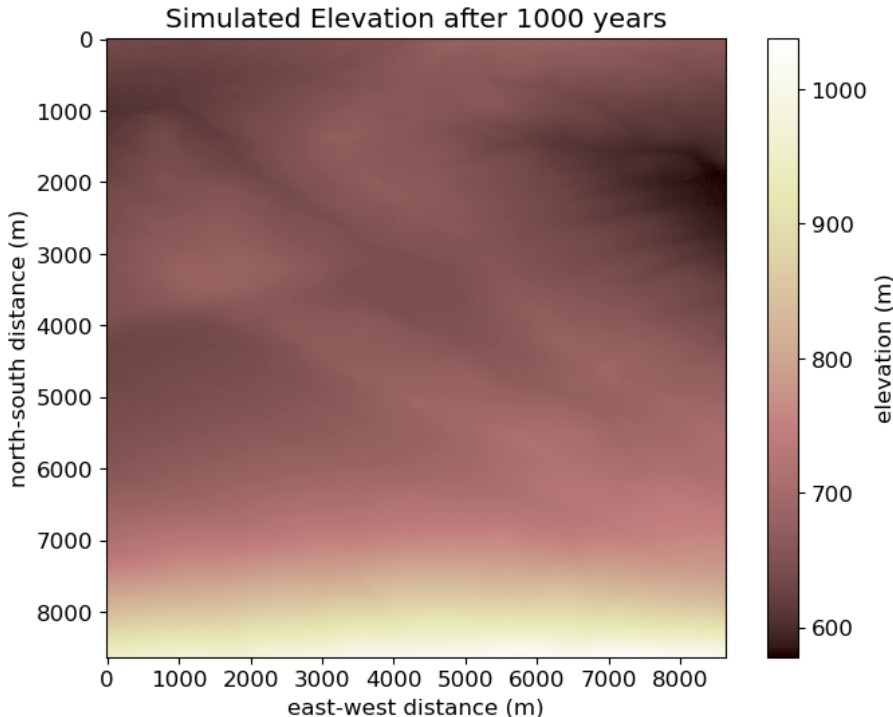

**Figure 9: Hillslope evolution result of the Landlab DepthDependentDiffuser component for the Eight Mile Lake area.**

### 3.2.4 Wave Power

Energetic waves cause shoreline erosion, change geomorphology, and generate renewable energy (Hansen and Barnard, 2010; Mwasilu and Jung, 2019; Vousdoukas et al., 2020). Globally, around 28,000 km$^2$ of otherwise-permanent coastal land was lost from 1984 to 2015, which is double the amount of land gained over this same period (Mentaschi et al., 2018). Wave power can be a useful predictor of shoreline change (e.g., beaches: Davidson et al., 2013; marshes: Leonardi et al., 2016), with higher wave heights and longer wave periods leading to larger wave power. Wave power is also used to assess feasibility of renewable energy generation (Ozkan and Mayo, 2019; Thorpe, 1999). This use case therefore focuses on extracting and analyzing wave characteristics and calculating wave power for the Louisiana Shelf in the Northern Gulf of Mexico.

The National Oceanic and Atmospheric Administration (NOAA) runs the WAVEWATCH III model (Booij et al., 1999) on several different grids (EMC Operational Wave Product Table, 2023). The WAVEWATCH III Data Component increases the accessibility and useability of model estimates and is used here to facilitate wave power calculations. WAVEWATCH III variables, including significant wave height, peak wave period, peak wave direction, and east-west and north-south wind speeds were downloaded using this Data Component. For the analysis, data from the Gulf of Mexico and Northwest Atlantic grid were used because of the relatively high resolution of 4 arcminutes (~7400 m at the study site). Data for the summer of 2005 was interpolated to a specific location (28.8°N, 276.4°E) on the Louisiana Shelf and shown in Fig. 10. For this figure, wave direction is given in meteorological convention, with

0 degrees meaning that waves are coming from the north and 90 degrees meaning waves are coming from the east. Winds are also given in meteorological convention, meaning positive v values are coming from the north and positive u values are coming from the east. Wave power was then calculated using the WAVEWATCH III estimates of significant wave height and peak wave period for this location. The result was visualized using a time series and a rose diagram (Fig. 11 and Fig. 12). Results indicate that significant wave height and therefore wave power were larger in mid-March through mid-April, compared to later portions of Spring 2005. Waves were primarily traveling northwestward, including during the time periods with larger wave power. This use case demonstrates how the WAVEWATCH III Data Component can be used to analyze wave conditions that are important for coastal shoreline change and renewable energy generation.


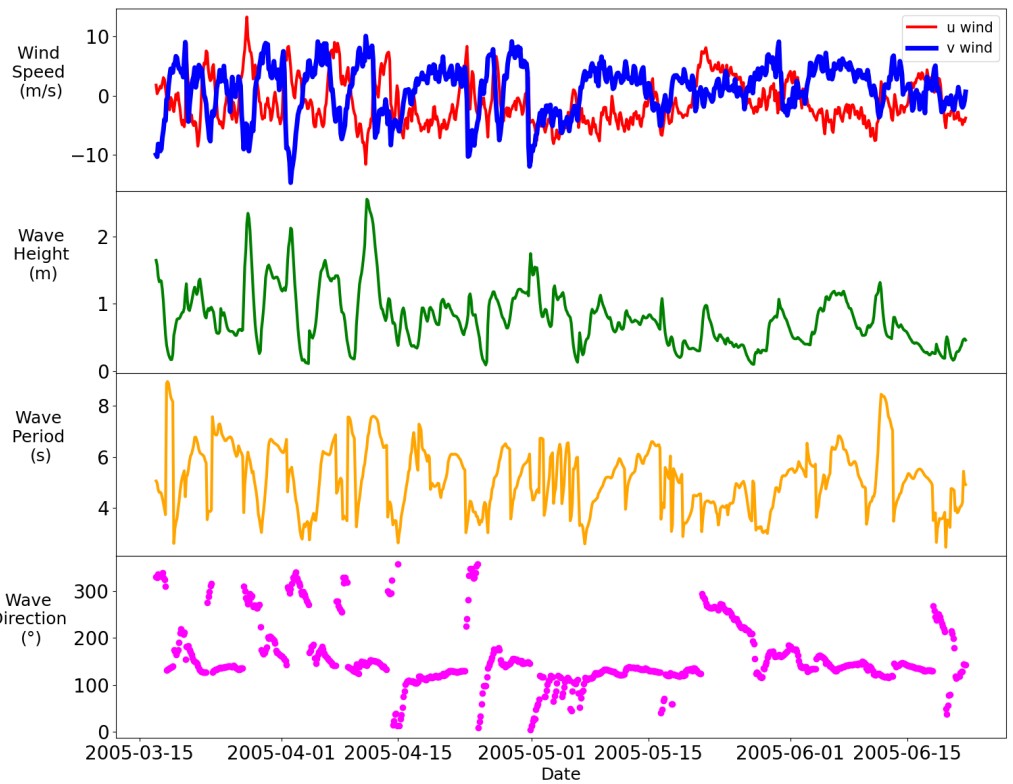


**Figure 10: Time series of the wave characteristics from WAVEWATCH III interpolated to 28.8°N, 276.4°E in the Gulf of Mexico.**


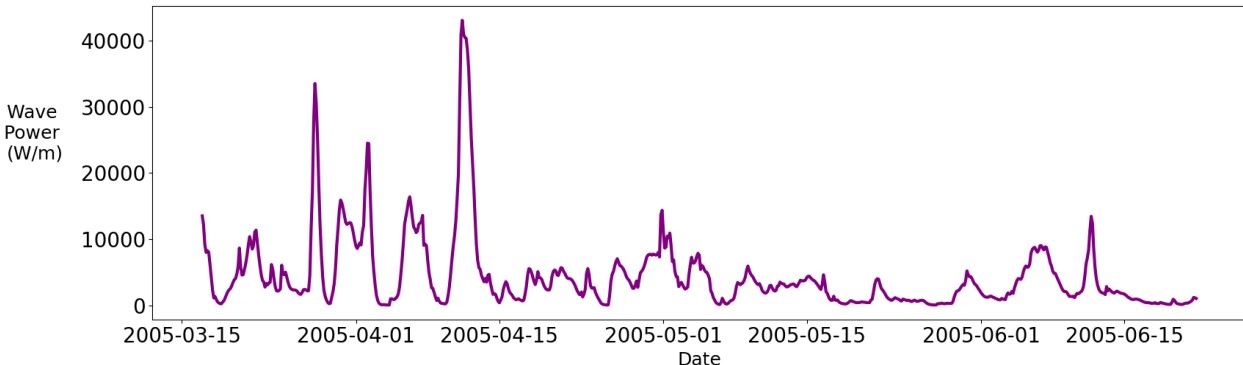

**Figure 11: Time series of wave power at 28.8°N, 276.4°E in the Gulf of Mexico.**

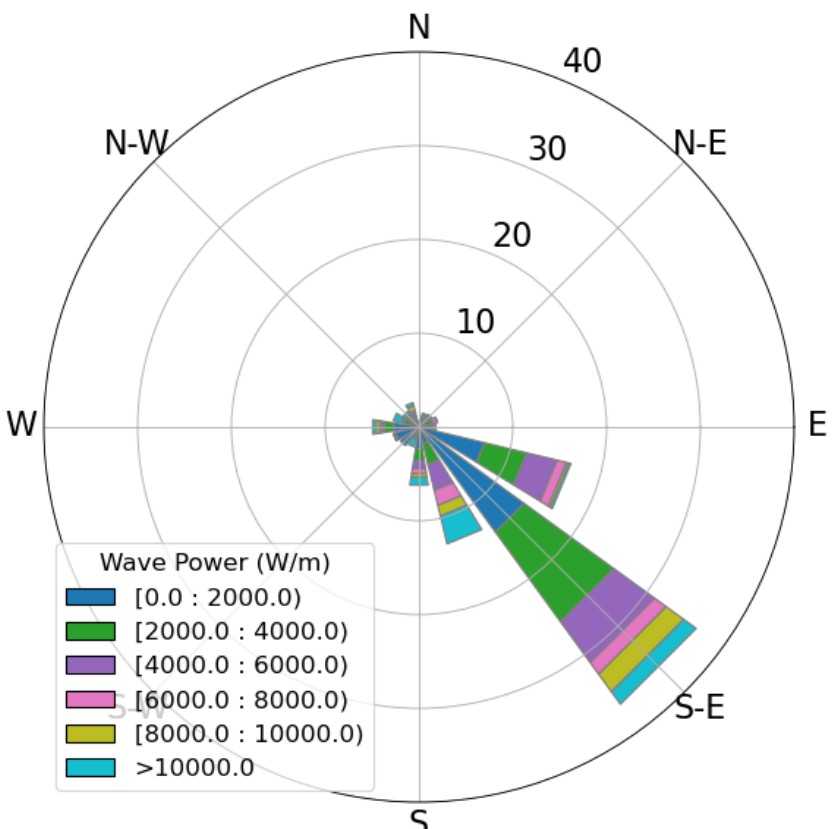

**Figure 12: Rose diagram of wave power at 28.8°N, 276.4°E in the Gulf of Mexico. The length of each bar and the concentric circles indicate the percentage of datapoints with waves coming from that direction (meteorological convention). The color indicates the wave power.**

### 3.3 Discussion

The case studies demonstrated that the Data Component design can be applied to a variety of datasets to support data-model integration for Earth surface processes research. These case studies also demonstrated multiple ways of using the Data Components. For example, the landsliding use case exemplifies how to use the Babelized component within the pymt modeling framework for data analysis. In Fig. 13, both the Topography and ERA5 Data Components are imported from the pymt module. Despite the different data sources and file formats for those Data Components, the

methods to initialize an instance and to access variables and grid information remains the same.

```python
from pymt.models import Topography, Era5
```

```python
# initialize Topography data component
dem = Topography()
dem.initialize(os.path.join(config_dir, 'dem_config.yaml'))

# get DEM variable info
var_name = dem.output_var_names[0]
var_unit = dem.var_units(var_name)
var_location = dem.var_location(var_name)
var_type = dem.var_type(var_name)
var_grid = dem.var_grid(var_name)
var_itemsize = dem.var_itemsize(var_name)
var_nbytes = dem.var_nbytes(var_name)
print('variable_name: {} \nvar_unit: {} \nvar_location: {} \nvar_type: {} \nvar_grid: {} \nvar_itemsize: {}'
            '\nvar_nbytes: {} \n'. format(var_name, var_unit, var_location, var_type, var_grid, var_itemsize, var_nbytes))

# get DEM grid info
dem_grid_ndim = dem.grid_ndim(var_grid)
dem_grid_type = dem.grid_type(var_grid)
dem_grid_shape = dem.grid_shape(var_grid)
dem_grid_spacing = dem.grid_spacing(var_grid)
dem_grid_origin = dem.grid_origin(var_grid)

print('grid_ndim: {} \ngrid_type: {} \ngrid_shape: {} \ngrid_spacing: {} \ngrid_origin: {}'.format(
    dem_grid_ndim, dem_grid_type, dem_grid_shape, dem_grid_spacing, dem_grid_origin))
```

(a)

```python
# initialize ERA5 data component
era5 = Era5()
era5.initialize(os.path.join(config_dir,'era5_config.yaml'))

# get ERA5 variable info
for var_name in era5.output_var_names:
    var_unit = era5.var_units(var_name)
    var_location = era5.var_location(var_name)
    var_type = era5.var_type(var_name)
    var_grid = era5.var_grid(var_name)
    var_itemsize = era5.var_itemsize(var_name)
    var_nbytes = era5.var_nbytes(var_name)
    print('variable_name: {} \nvar_unit: {} \nvar_location: {} \nvar_type: {} \nvar_grid: {} \nvar_itemsize: {}'
            '\nvar_nbytes: {} \n'. format(var_name, var_unit, var_location, var_type, var_grid, var_itemsize, var_nbytes))

# get ERA5 grid info
era5_grid_ndim = era5.grid_ndim(var_grid)
era5_grid_type = era5.grid_type(var_grid)
era5_grid_shape = era5.grid_shape(var_grid)
era5_grid_spacing = era5.grid_spacing(var_grid)
era5_grid_origin = era5.grid_origin(var_grid)

print('grid_ndim: {} \ngrid_type: {} \ngrid_shape: {} \ngrid_spacing: {} \ngrid_origin: {}'.format(
    era5_grid_ndim, era5_grid_type, era5_grid_shape, era5_grid_spacing, era5_grid_origin))
```

(b)

**Figure 13: Scripts from the landslide use case to demonstrate using the Topography Data Component (a) and the ERA5 Data Component (b) within pymt.**


The rainfall-runoff modeling use case demonstrates the ability to combine a Data Component (the Topography Data Component) with a Landlab grid object and a Landlab Model Component (the FlowAccumulator) (Fig. 14). The key aspect of this process involves defining an instance of the RasterModelGrid ('model_grid') based on the features of the Data Component ('dem'). Subsequently, this model grid is passed as a parameter to create an instance of a Model

Component ('fa'), which links the data and the computational aspects of the modeling process.

```python
# get DEM variable data
dem_data = dem.get_value(var_name)

# set up raster model grid
model_grid = RasterModelGrid(dem_grid_shape, xy_spacing=30)

# add topographic elevation data field
dem_field = model_grid.add_field("topographic__elevation", dem_data.astype('float'))

# calculate the flow accumulation
fa=FlowAccumulator( model_grid, method='Steepest',
                    flow_director='FlowDirectorSteepest',
                    depression_finder='LakeMapperBarnes',
                    redirect_flow_steepest_descent=True,
                    reaccumulate_flow=True)
fa.run_one_step()
```

**Figure 14: Scripts from the rainfall-runoff modeling use case to demonstrate coupling the Topography Data Component with FlowAccumulator Component from Landlab.**


The permafrost thaw and hillslope diffusion use case demonstrates pre-processing datasets using a Data Component and feeding the resulting data as inputs to a pymt Model Component. The example demonstrates how a simple and compact code can retrieve time series data for a given study area from the Data Component ('era5' and 'era5_2') (Fig. 15 (a)), and use this to set up and run the Kudryavtsev model using the prepared inputs ('input_data') (Fig. 15 (b)).

Notably, within the pymt modeling framework, the methods to create an instance ('initialize()'), to retrieve data values from the component ('get_value()'), and to update the time step ('update()') remain consistent for both the Data and Model Components.

```python
# create dataframe to store time series data
era5_df = pd.DataFrame(columns = ['temp','swe','dens','time'])
time_steps = 12*10  # 10 years of monthly data

for data_comp in [era5, era5_2]:

    for i in range(0, time_steps):
        # get values
        temp = data_comp.get_value('2 metre temperature')
        swe = data_comp.get_value('Snow depth')
        dens = data_comp.get_value('Snow density')
        time = cftime.num2pydate(data_comp.time, data_comp.time_units)

        # add new row to dataframe
        era5_df.loc[len(era5_df)]=[temp[0], swe[0], dens[0], time]

        # update to next time step
        data_comp.update()

era5_df = era5_df.set_index('time')
```

(a)

```python
# setup model
ku = Ku()
args = ku.setup(start_year=start, end_year=end, lat=63.88, lon=-149.25)
ku.initialize(*args)

# run model
for index, row in input_data.iterrows():
    ku.set_value("atmosphere_bottom_air__temperature", row['temp_mean'])
    ku.set_value("atmosphere_bottom_air__temperature_amplitude", row['temp_amp'])
    ku.set_value("snowpack__depth", row['snow_h'])
    ku.update()

    # store result
    active_layer.loc[index] = ku.get_value('soil__active_layer_thickness')[0]
```

(b)

**Figure 15: Scripts from the permafrost thaw and hillslope diffusion use case. Panel (a) shows retrieving time series data from ERA5 Data Component; (b) shows the Kudryavtsev model simulation.**

The wave power use case demonstrates the use of the API available within the BMI component for data access instead of using the Babelized component. This approach becomes advantageous particularly when there is no need to couple the Data and Model Components for analysis. In Fig. 16, the API ('WaveWatch3') for downloading the WAVEWATCH III datasets is imported from the BMI component ('bmi_wavewtch3'). This API provides methods that extend beyond the standard BMI methods. For instance, the 'inc' method allows users to access additional months of data without needing to create new instances of the Data Component for each month, which simplifies the data retrieval process.

```
# Load in the bmi-wavewatch3 data component
from bmi_wavewatch3 import WaveWatch3

# Specify the time period and the coordinates of interest

# Starting month
start_month = "2005-03-01"

# Number of months after to pull
num_months = 3

# Start date (specific date to start data)
start_date = "2005-03-17"

# End date (specific date to end data)
end_date = "2005-06-22"

# Specify the grid
grid = 'at_4m' # 'at_4m' = Atlantic grid at 4 arcminute resolution; see figure in background section

# Specify the lat lon we want (the one point)
lat = 28.8 # degrees
lon = 267.4 # degrees

# Fetch the data for the time period we want (to start at) and the grid we want
ww3 = WaveWatch3(start_month, grid=grid)

# Save the data to a list
months = [ww3.data]

# Print info about the data
ww3.data

# Add on the additional months
for _ in range(num_months):
    ww3.inc()
    months.append(ww3.data)
```


**Figure 16: Scripts from the wave power use case to demonstrate using the API in the BMI component for data access.**

From the implementation and use cases of the Data Components, we found that our design provides benefits in the following aspects. 1) Usability: since the datasets are wrapped with BMI, the methods to get metadata and data values are the same regardless of their file formats or the grid types. This feature can be seen in the four use cases, where the code for retrieving the variable and grid information is the same across a wide range of data types and file formats. This simplifies the process of learning about new Data Components for users who are already familiar with the basic design. Additionally, because the Model Components also adopt the BMI methods, it becomes intuitive for users to know how to couple the data and model components together. 2) Reproducibility: Data Components are implemented as open-source Python packages, which enables users to document the data-model integration workflows in the Jupyter Notebooks for tracking and sharing computational analysis. Compared with the modeling frameworks that allow users to create modeling workflows via GUIs, the Data Component design helps to provide detailed information for data access and preparation behind the scenes. 3) Flexibility: the design provides a flexible way of using Data Components. Users can either use the API directly for data analysis when there is no need to couple data with models (as with the wave power use case) or use the Babelized component under the modeling framework (as exemplified by the rainfall-runoff modeling use case), which can make it easier to write efficient code for different situations. In addition, this design provides the flexibility to make the Data Components work within any modeling frameworks or tools that

support, or are compatible with, the BMI standard (e.g., Landlab) without making additional changes to the Data Components.

While developing the use cases, we also identified the limitations of the existing BMI methods to represent the features of datasets. For instance, there is a need to add new methods to access the spatial reference information of the datasets, which can facilitate data reprojection and regridding to convert heterogeneous datasets to the same grid resolution and coordinate system. Moreover, the existing BMI methods mainly support wrapping datasets with spatial and time dimensions, and it becomes challenging to deal with datasets that include dimensions representing other variables. Take the ERA5 datasets as an example: there are ensemble model simulation results that include dimensions representing the ensemble number and/or the pressure levels. The existing BMI methods don't support accessing the information for those types of dimensions, so the current implementation of the ERA5 Data Component mainly supports datasets that only include spatial and time dimensions. This highlights a need for extensions to the core BMI standard that can accommodate these needs and enhance the usability of the Data Components.

Currently, new Data Component and use cases are also under development. One example is the ROMS Data Component designed to access the model outputs of the Regional Ocean Modeling System (ROMS) (Haidvogel et al., 2008). The ROMS Data Component will be coupled with the Landlab and pymt Model Components to help explore the fate of particulate organic carbon in the Arctic, including its release via permafrost thaw, transport and oxidation in the fluvial and coastal systems, and its burial in offshore sediments. Data Components are designed to serve as open resources by and for the science community, and we highly encourage readers to develop and share their own Data Components.

## 4 Conclusions

The integration of data and numerical models plays a vital role in advancing the understanding of the complex processes of Earth systems. However, with the increasing number of datasets available on the internet and the growing trend of reproducible computational research, there is a need to provide a convenient and standardized way to access a variety of datasets and easily couple them with diverse models to improve the efficiency and reproducibility of the data-model integration workflows.

This paper presents an approach that uses open-source software and standards from the CSDMS Workbench to create 'Data Components' that support open data-model integration for Earth surface processes modeling. A Data Component is a dataset wrapped with BMI functions. To evaluate and illustrate our approach, we implemented several Data Components for datasets in various file formats and grid types, and then applied them in research demonstrations related to landsliding, overland flow, permafrost, and ocean waves. The results demonstrated that the Data Component design provides a consistent way to access and use online datasets from multiple sources and to easily couple data with models, which increases the accessibility and reusability of research datasets.

Another advantage of the Data Component design is that it enables researchers to document the data-model integration workflow in a Jupyter Notebook or similar 'literate programming' format (Knuth, 1984), which helps other researchers to discover, access, operate, and reuse computational research through online platforms. This approach can help

improve research transparency and workflow reproducibility to encourage collaboration. Moreover, our use cases can
be adapted and applied to other study sites so that researchers can rapidly set up modeling studies after or during a
geophysical event to have a quick exploration or initial assessment of the associated hazards. Although our case studies
are centered on Earth surface processes and natural hazard impacts, the core concepts of the Data Component design
are extensible to datasets in other scientific domains.

In the future, we will focus on developing new Data Components and extending BMI to support a wider range of
datasets. We will also provide educational materials to encourage the geoscience community to apply existing, or
implement new, Data Components to create reproducible data-model integration workflows.

**Code Availability**

**NWIS Data Component:**
BMI component: https://doi.org/10.5281/zenodo.10368806
pymt plugin:  https://doi.org/10.5281/zenodo.10368876
**Topography Data Component:**
BMI component: https://doi.org/10.5281/zenodo.8327417
pymt plugin: https://doi.org/10.5281/zenodo.10308417
**SoilGrids Data Component:**
BMI component: https://doi.org/10.5281/zenodo.10368883
pymt plugin: https://doi.org/10.5281/zenodo.10368885
**ERA5 Data Component:**
BMI component: https://doi.org/10.5281/zenodo.10368879
pymt plugin: https://doi.org/10.5281/zenodo.10368881
**WAVEWATCH III Data Component:**
BMI component: https://doi.org/10.5281/zenodo.8326599
**Use Case Jupyter Notebooks**:
https://doi.org/10.4211/hs.28af99c09ee4423dbffef28bf32837e0

**Author's Contribution**

Mark Piper, Eric Hutton, and Tian Gan developed the Data Components. Tian Gan, Benjamin Campforts, Brianna
Undzis, Ethan Pierce, Greg Tucker, Irina Overeem, and Julia Moriarty created the use case Jupyter Notebooks. Tian
Gan prepared the manuscript draft and all co-authors reviewed and edited the manuscript.

## Competing interests

The authors declare that they have no conflict of interest.

## Acknowledgements

This work was supported by the National Science Foundation under collaborative grants 1831623, 2026951, 2104102, and 2148762. Any opinions, findings, and conclusions or recommendations expressed in this material are those of the authors and do not necessarily reflect the views of the National Science Foundation. We also thank our collaborator

Dr. Anthony Castronova for his technical support of the CUAHSI JupyterHub.

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
