# Peer review of "CSDMS Data Components: data-model integration tools for Earth surface processes modeling"

_Geoscientific Model Development, 2023_

## Author Comment (AC1)

**Response to Reviews**

**gmd-2023-127: CSDMS Data Components: data-model integration tools for Earth surface processes modeling**

Dec, 2023

**Dear Reviewers and Editors,**

We are very thankful for the constructive and valuable feedback on our manuscript. We considered all the comments and made changes to reflect most of the suggestions from the reviews. Our responses are provided below highlighted in blue text.

**Reviewer #1**

General comments:

This paper describes a new set of data model components that allow accessing a variety of data sources via a uniform interface. This is a valuable approach to make it easier for model developers and users to access forcing data for their modeling systems without needing to learn a new interface or write substantial amounts of code to access data for each new data source. These data models are part of the Community Surface Dynamic Modeling System (CSDMS). Not having familiarity with the CSDMS modeling system, it's hard for me to speak to how much use this will get, but these new data model components do seem useful. I also enjoyed taking this opportunity to learn a little bit about CSDMS, which seems like a valuable project for the community.

The paper is organized well and is well-written. I appreciate the inclusion of a code snippet (Figure 2), illustrating how these data components can be used in practice. The associated web page (https://csdms.colorado.edu/wiki/DataComponents) gives a catalog of existing data models and instructions for creating a new data model, which is also very useful.

Response: Thank you very much for recognizing the contribution of this work and giving us useful comments. Based on your suggestions, we added more details to clarify what current functions can do and how they work based on your comments. We also added text and figures for the use cases to describe the mechanics of using the Data Components. For more details, please check our response for each comment.

Specific comments:

1. Many of the data components involve downloading data from a remote server. Can the authors include some more details on the logic for this? Particularly for time-varying data, I wasn't clear on whether the full time series is downloaded at once – e.g., in the initialize step – or if each time sample is downloaded on an as-needed basis in the run

loop. If the full time series is downloaded in initialization, is there a way to specify just a subset of the full time series for downloading? If each time sample is downloaded on an as-needed basis, is there a significant performance impact of these repeated remote accesses? The authors mention caching of the data (line 169); can they elaborate a bit on how this works? Does this mean that, if you run the same data component multiple times on the same machine, the download will only happen the first time, or is the caching only within a single model run?

Response: We added descriptions in the manuscript to include more details on how the Data Component works in Section 3.1. For time-varying data, the full time series is downloaded at once in the initialize step. Take the NWIS Data Component as an example, if the user wants to get the discharge time series for 2023-1-1 to 2023-2-1, the "initialize" function will download the data for the specified time at once and save the time series in the NetCDF file. As for the caching of the data, if the user runs the same Data Component (with the same configuration file) multiple times on the same machine, the download will only happen the first time.

2. It is helpful that the authors included some use cases (section 3.2) to provide some example usage of these data components. However, I didn't find much value after the first two: The additional use cases didn't really help me gain any better understanding of how these data model components are used. (This is not to say that the first two were necessarily the most valuable: I just think that, after reading about two different use cases, I didn't get much out of the additional use cases.) I recommend that the authors do one or both of these:

   o Cut one or two of these use cases

   o Rework the description of these use cases to focus more on the mechanics of using the data components and less on other aspects of the specific scientific study. At the end of each use case, the authors briefly mention what is interesting about the use case, such as (Lines 433-434), "demonstrates how to use the API of the Data Component instead of the Babelized component for data access, when there is no need to couple Data and Model Components for analysis". But it's hard for me to get a clear picture of how this works based on the given descriptions. I would be interested in more focus on the mechanics of the data model use, through schematic figures and/or code snippets (actual code or pseudocode), and less focus on the scientific results (which is not the main point of this paper).

Response: Thank you for the feedback and suggestions! The reason we provided four use cases is that we hope to help people from different domain fields get an idea for how the Data Components can be used in various research topics. Thus, we decided to keep the four use cases in Section 3.2 to focus on the description of the modeling workflow. We also added new text and figures in Section 3.3 to put more focus on the mechanics of using the Data Components.

3. The authors may also be interested in the Community Data Models for Earth Predictive Systems (CDEPS; see https://github.com/ESCOMP/CDEPS and https://escomp.github.io/CDEPS/versions/master/html/index.html). While these are

aimed more specifically at Earth system models that are coupled using ESMF, their concept is somewhat similar to the data components described here. (I leave it up to the authors whether they think this is worth citing.)

Response: Thank you for pointing out this research work. We added new description in Section 1 to mention about this work in our manuscript.

Technical corrections

This paper is generally very well written, but there are a few minor grammatical / technical errors. Ones I noticed are:

1.  Line 66: "and together with" should be "together with"

    Response: We updated the text based on your suggestion.

2.  Line 176: Change "And the Babelizer will use" to "The Babelizer will then use"

    Response: We updated the text based on your suggestion.

3.  Figure 2b: There is a vertical bar at the end of the line setting var_grid which appears to be the cursor; this is a bit confusing since it could be interpreted as a vertical bar character

    Response: We updated the figure to remove the cursor.

4.  Line 504: "support wider range" should be "support a wider range"

    Response: We updated the text based on your suggestion.

**Reviewer #2**

Synopsis:

In this manuscript, Gan and coauthors outline a novel piece of cyber infrastructure that they have devised to support data/model assimilation for Earth-surface processes research. They demonstrate the overall structure of their cyber infrastructure they place it in the context of previous work at CSDMS. They demonstrate its scientific value with four case studies.

Overall Comments:

I found this manuscript nice to read, easy to understand, and broad in its impact and scope. I think that the major components that the authors highlight here have been well explained. The authors compellingly demonstrate how their new tool will accelerate geoscience research.

After viewing some of the example notebooks, and attempting to set them up myself on a local machine, I ran into a few installation and software problems. I think these problems are simply software bugs and may be related to a changing API. They are not important to resolve here, and are best addressed through github issues.

Response: Thank you very much for your positive feedback on our manuscript! We really appreciate that you tried to test those example notebooks. Based on your feedback, we ran the notebooks on the online platforms and the local PC to make sure they are working as expected.

I do have a few (very) minor language suggestions:

lines [468-477] it's not totally clear to me in this section what parts have been implemented, and what parts Remain the implemented. For example, you state that it "will improve the usability of data components by extending the BMI standard". Have you done so? Or does that remain to be done? It would be nice to just be completely explicit here; the language is a little bit ambiguous.

Response: We plan to extend the BMI standard in the future and the work is not completed yet. We modified the sentences in the manuscript to clarify the information.

Lines [354-356] here I think the sentence that begins with " with the trend of global warming..." needs a little bit of a rewrite, there are some mismatches between subject and verb that make it hard to parse.

Response: Thanks for pointing this out. We modified the sentences to make it easy to understand.

In any case, I recommend that it be published.

I encourage the authors to get in contact with me if they have any questions about this review.

eric.barefoot@ucr.edu

**Executive Editor**

Dear authors,

in my role as Executive editor of GMD, I would like to bring to your attention our Editorial version 1.2: https://www.geosci-model-dev.net/12/2215/2019/

This highlights some requirements of papers published in GMD, which is also available on the GMD website in the 'Manuscript Types' section: http://www.geoscientific-model-development.net/submission/manuscript_types.html

In particular, please note that for your paper, the following requirement has not been met in the Discussions paper:

Code must be published on a persistent public archive with a unique identifier for the exact model version described in the paper or uploaded to the supplement, unless this is impossible for reasons beyond the control of authors. All papers must include a section, at the end of the paper, entitled "Code availability". Here, either instructions for obtaining the code, or the reasons why the code is not available should be clearly stated. It is preferred for the code to be uploaded as a supplement or to be made available at a data repository with an associated DOI (digital object identifier) for the exact model version described in the paper. Alternatively, for established models, there may be an existing means of accessing the code through a particular system. In this case, there must exist a means of permanently accessing the precise model version described in the paper. In some cases, authors may prefer to put models on their own website, or to act as a point of contact for obtaining the code. Given the impermanence of websites and email addresses, this is not encouraged, and authors should consider improving the availability with a more permanent arrangement. Making code available through personal websites or via email contact to the authors is not sufficient. After the paper is accepted the model archive should be updated to include a link to the GMD paper.

As GitHub is not a persistent archive, please provide a persistent release for the exact data components as discussion in this publication in this paper. As explained in https://www.geoscientific-model-development.net/about/manuscript\_types.html the preferred reference to this release is through the use of a DOI which then can be cited in the paper. For projects in GitHub a DOI for a released code version can easily be created using Zenodo, see https://guides.github.com/activities/citable-code/ for details.

Please note, best practice is to publish both, the URL for the updated repositories and the permanently archived version of the code / data used for this publication.

Response: Thank you for the information and suggestions! We published the Data Components using Zenodo and the use case notebooks using HydroShare. We cited these research products in our manuscript. The "Reference" and "Code availability" sections are also updated.